# Left Anterior Sectorectomy: An Alternative to Left Hepatectomy for Tumors Invading the Distal Part of the Left Portal Vein

**DOI:** 10.3390/diagnostics12020545

**Published:** 2022-02-21

**Authors:** Mattia Garancini, Mauro Alessandro Scotti, Luca Gianotti, Cristina Ciulli, Francesca Carissimi, Fabio Uggeri, Luca Degrate, Marco Braga, Fabrizio Romano

**Affiliations:** Unit of Hepatobiliopancreatic Surgery, Department of General Surgery I, ASST-Monza, San Gerardo Hospital, University of Milano-Bicocca, 20900 Monza, Italy; mauro.scotti@asst-monza.it (M.A.S.); luca.gianotti@unimib.it (L.G.); ciulli-300670@asst-monza.it (C.C.); f.carissimi@campus.unimib.it (F.C.); fabio.uggeri@unimib.it (F.U.); l.degrate@asst-monza.it (L.D.); marco.braga@unimib.it (M.B.); fabrizio.romano@unimib.it (F.R.)

**Keywords:** liver resection, left anterior sector, macrovascular invasion, parenchima sparing, intra-operative ultrasound

## Abstract

Background: Liver tumors invading the distal part of the umbilical portion of the left portal vein usually require left hepatectomy. The recent introduction of the concept of left anterior sector, an independent anatomo-functional unit including the anterior portion of the left liver and supplied by the distal part of the umbilical portion of the left portal vein, could represent the rational for an alternative surgical approach. The aim of this study was to introduce the novel surgical procedure of ultrasound-guided left anterior sectorectomy. Methods: Among 92 consecutive patients who underwent hepatectomy, 3 patients with tumor invading the distal part of the umbilical portion of the left portal (two with colorectal liver metastases and one with neuroendocrine tumor liver metastases) underwent left anterior sectorectomy alone or in association with liver multiple metastasectomies. Results: Mean operation time was 393 min; post-operative morbidity and mortality were not observed. After a mean FU of 23 months (range 19–28), no local recurrence occurred. Conclusions: In presence of tumors invading the distal part of the umbilical portion of the left portal, left anterior sectorectomy could be considered as an anatomic radical surgical option that is safe but more conservative than a left hepatectomy.

## 1. Introduction

The surgical treatment of tumors invading the distal part of the umbilical portion of the left portal vein (UPLPV) may represent a challenge for liver surgeons (Figure 1a). The presence of tumors with certain or suspected invasion of the UPLPV, when suitable for surgery, requires a radical anatomical liver resection with removal of the tumor, the invaded vessel, and the portion of parenchima supplied by that vessel; the traditional radical surgical approach in such cases is represented by a major resection, the left hemi-hepatectomy (Figure 1b).

However, a modern approach in liver surgery needs to balance the oncological radicality with the aim to minimize the volume of liver to be resected [1,2]. When planning a hepatic resection, a parenchima-sparing policy is often mandatory for several reasons: first, to reduce post-operative morbidity and mortality, and second, in the perspective of expansion of surgical indications, in particular among patients with poor liver function (as cirrhotic patients) or with multiple bilobar metastases [3,4,5,6].

With the aim to identify alternative and more conservative surgical procedures for the treatment of patients with tumors invading the UPLPV, we recently performed an anatomo/radiological study and introduced the novel anatomical concept of left anterior sector (LAS), an independent anatomo/functional unit corresponding to the anterior portion of the left hemiliver, including segment 3 (S3) and S4 inferior (S4b), and supplied by an independent sectorial portal pedicle (PP) represented by the distal part of the UPLPV downstream from the origin of independent PPs for left posterior segments (S2 and S4a) [7] (Figure 1c). The definition of this new anatomical entity assumed that S4 should be divided into two separated sub-segments; this was based on the finding that, in 79.5% of patients, S4a and S4b are supplied by independent portal pedicles (PPs) and should be considered two separated anatomo-functional units [7].

This novel concept of LAS appeared to be consistent with the formal definition of hepatic sectors as postulated by Bismuth H.: the portion of parenchyma individualized by hepatic veins and supplied by an independent sectorial PP [8]; besides, this categorization of the left liver in terms of anterior/posterior appeared to be anatomically faithful and was considered relevant to open the field to new surgical approaches in patients with tumors invading the distal part of the UPLPV.

The aim of the present study was to introduce the surgical procedure of removal of the LAS for presence of tumor invading the distal part of the UPLPV: the left anterior sectorectomy (Figure 1d).

## 2. Materials and Methods

Among 92 consecutive patients with primitive or metastatic liver tumors who underwent liver resection from 1 May 2019 to 31 December 2020 in the unit of hepatobiliopancreatic surgery, Department of General Surgery 1, San Gerardo Hospital, ASST-Monza, Italy, 3 patients underwent left anterior sectorectomy for presence of tumors in the LAS invading the distal part of the UPLPV.

For patients who underwent left anterior sectorectomy, the following parameters were collected in an anonymized database and analyzed: age; gender; number and site of liver’s tumors; pattern of LPV’s branching (following the classification previously reported [7]); radiological pre-operative and intra-operative confirmation of presence of LAS; operative time; intra-operative blood loss; Pringle time; types of surgical procedures; post-operative complication rate; mortality rate; histological features; follow up (FU); and local recurrence.

### 2.1. From the LAS to Left Anterior Sectorectomy: The Indications to Surgery

The careful pre-operative evaluation of Contrast-Enhanced Computed Tomography (CT) was essential before the surgical indication to left anterior sectorectomy; CT allowed to perform the detection of liver lesions and to ascertain the relationship between the liver lesions and the left portal system. Moreover, the CT was used to study the anatomy of the LPV and to identify the pattern of LPV’s branching following the classification previously reported: normal anatomy with at least 1 independent PP for each of S2, S3, S4a, and S4b (pattern I, 74.5%); anomaly with only 1 common PP supplying S2/S3 (pattern II, 5%); anomaly with only 1 common PP for S4a/S4b (pattern III, 18.5%); and anomaly with 1 common PPs for both S2/S3 and S4a/S4b (pattern IV, 2%) [7].

The identification of the pattern of LPV’s branching was fundamental in order to radiologically define the presence of LAS (typically present in pattern I patients but identifiable even in patterns II and III patients, overall present to 98% of cases) and to define the adequate level of section of the sectorial PP of the LAS represented by the UPLPV [7] (Figure 2).

The evaluation of the left hepatic artery’s branching and of the left liver’s bile duct system were invaluable before planning a left anterior sectorectomy; the arterial anatomy was studied in the arterial phase of the CT, while the biliary tree was pre-operatively studied with a Magnetic Resonance Imaging (MRI) with liver-specific contrast agent.

### 2.2. Left Anterior Sectorectomy: The Surgical Procedure

A J-shaped laparotomy was performed. Round, falciform, left triangular, and left coronary ligaments were sectioned. Intra-operative ultrasound (IOUS) was performed in order to confirm the pre-operative findings, the indication, and the technical feasibility of the surgical procedure. Definitely, presence of LAS was confirmed only in cases in which the theoretical section of the UPLPV and removal of S3 and S4b was considered possible without damaging S2 and/or S4a and their segmental PPs.

The precise section level of the sectorial PP was identified between the origins of segmental PPs for the anterior (S3 and S4b) and posterior (S2 and S4a) segments with ultrasound-guided technique. The boundaries of LAS were delimited: medial and lateral landmarks were the middle hepatic vein (MHV) and the left hepatic vein (LHV). In order to identify the intersegmental planes between S2/S3 and between S4a/S4b, transient ischemia by means of surgical isolation plus closure with a bulldog clamp or compression technique [9] of segmental PPs of S3 and/or S4b at their origin from UPLPV were used. The transient ischemia showed the cranial boundary of the LAS corresponding to the intersegmental planes between the origin and course of the PPs supplying left anterior segments (S3, S4b) and the origin and course of independent PPs supplying the posterior segments (S2, S4a). In presence of tumor’s invasion of the origin of PPs for S3 or S4b, blunt compression technique was applied directly on the UPLPV at the level of the planned site of section; otherwise, the intersegmental plane could be identified with IOUS in a plane between the tumor and the origin and course of proximal independent portal pedicle for S2 or S4a. The delimitation of the LAS was drawn with electrocautery on the liver surface (Figure 3A).

The relationship between the LAS’ plane and the branching of the left hepatic artery and biliary duct were studied with IOUS in order to avoid the section of arterial and biliary structures supplying the left posterior segments (S2 and S4a) during the left anterior sectorectomy. The left hepatic artery usually branches in a vessel feeding S4 and a vessel feeding the left lobe; the same branching is usually present for the left hepatic bile duct. The LAS is a peripheral portion of liver; in this limited series, the transection plane of the LAS was always more peripheral than these bifurcations of left hepatic artery and bile duct, and the arteries and bile ducts for S3 and S4b were sectioned at the level of segmental/sub-segmental branches without impairment for the spared segments (S2 and S4a).

Parenchymal transection was performed using Cavitron Ultrasonic Surgical Aspirator (CUSA^®^, Integra LifeSciences. Plainsboro, NJ, USA); vessel coagulation was obtained using Aquamantys™ (Medtronic, Minneapolis, MN, USA) and bipolar electrocautery. Each vessel thicker than 2 mm was ligated with thin (3/4-0) sutures and/or clipped. Intermittent Pringle maneuver was used only in cases of significant bleeding.

The parenchymal transection was accomplished without sectioning any segmental PP until the skeletonization of the sectorial Glissonean pedicle of LAS was obtained (Figure 3B). This sectorial Glissonean pedicle included the UPLPV together with arterial and biliary branches for the LAS; in this limited series, the arteries and bile ducts within the sectorial Glissonean pedicle for the LAS were always represented by the peripheral portion of the artery/bile duct for S3 (on the left side of the sectorial PP) and the peripheral portion of the artery/bile duct for S4b (on the right side of the sectorial PP) separately.

Then, the sectorial Glissonean pedicle for the LAS was closed with an atraumatic vascular forceps. S2 and S4a were evaluated to ascertain the absence of congested or ischemic area and to assess the presence of proper inflow and outflow at the IOUS color-doppler study. Then, the Glissonean pedicle was sectioned and sutured with a double running suture or divided with a vascular strapler (Figure 3C,D). The cut surface of the liver was secured by 3/4-0 sutures, electrocautery, and hemostatic matrix (Floseal^®^, Baxter International, Deerfield, IL, USA). Closed 19 drains were always inserted.

When planning a left anterior sectorectomy, the cholecystectomy is not strictly necessary, but it is suggested; in this limited series, the cholecystectomy was always performed.

## 3. Results

Three patients with tumors invading the distal part of the UPLPV and who had undergone left anterior sectorectomy were retrospectively identified and enrolled in the study (in one case, the tumor invaded the UPLPV and the origin of PP for S3, and in two cases, the tumor invaded the UPLPV and the origin of the PP for S4b). In detail, the study enrolled two patients with multiple bilobar colorectal liver metastases (CRLM) who had undergone left anterior sectorectomy + multiple metastasectomy + adhesiolysis + cholecystectomy and one patient with bifocal neuroendocrine tumor (NET) liver metastases who had undergone left anterior sectorectomy + adhesiolysis + cholecystectomy. In one case, the surgical procedure was a re-operation after previous multiple metastasectomy for CRLM: such patient was initially considered borderline-resectable by means of a one-stage hepatectomy consisting in left hemi-hepatectomy + multiple metastasectomy because of minimal residual liver volume assessed by volumetric analysis of CT-scan; the planned left anterior sectorectomy allowed to avoid a major resection and offered an adequate residual liver volume with a one-stage procedure. All the procedures were open liver resection.

Patients included two females and one male; mean age was 64 years.

Regarding the surgical outcomes, mean operation time was 393 min, and post-operative morbidity and mortality were not observed. At definitive histological evaluation, the portal macrovascular invasion was confirmed in each case, and all the specimens were R0 resections.

After a mean FU of 23 months (range 19–28), no patient had cut-edge recurrence; in particular, no cut-edge recurrence in the site of left anterior sectorectomy occurred.

During the FU, both patients with CRLM had isolated liver recurrence after surgery; they were treated with chemotherapy. The patient with NET metastases had isolated peritoneal recurrence (without liver recurrence) that was treated with somatostatin. Patients’ characteristics and post-operative data are reported in Table 1.

## 4. Discussion

Better knowledge of liver anatomy may have a great impact on the surgical strategy and quality of hepatectomy [10,11,12].

The main point of discussion of the present study regards the introduction of a new surgical procedure based on a new anatomical entity. Following the Brisbane 2000 Nomenclature of Hepatic Anatomy and Resections (B2000) [13], the left liver (basing on the portal vein’s branching) should be divided in a left lateral sector (LLS) including S2 and a left medial sector (LMS) including S3 and S4; in detail, the portal branch for S2 should represent the sectorial PP for the LLS, and the UPLPV (downstream from the origin of the portal branch for S2) should represent the sectorial PP for the LMS. Such subdivision of the left liver is based on a description of S4 as an undivided segment in which both S4a and S4b are supplied by multiple portal branches originating from the right horn at the junction between the left portal vein and the round ligament (Figure 4a). With the aim to avoid a major resection and following the B2000, the above-described oncological situation (tumor invading the distal part of the UPLPV) could be theoretically surgically managed by means of a left medial sectorectomy (removal of S3 and S4, sparing S2), with section of the left medial sectorial PP downstream from the origin of the portal branch for S2 (Figure 4a). In this perspective of the B2000, the concept of LAS and the procedure of left anterior sectorectomy could appear redundant or useless.

On the other hand, several authors have recently described that 75–80% of patients have portal branches supplying the superior portion of S4 arising directly from the angle or the transverse portion of the LPV (TPLPV) [7,11,14,15]; in 67% of patients, such PPs represent an independent supply of S4a and originate at the same level or upstream from the origin of the sectorial PP for LLS [7]. The presence of these independent proximal portal branches for S4a represent the rational for considering S4a an independent anatomo-functional unit and has two important implications.

First, since in 67% of cases, the origin of the sectorial PP for the LLS arises among the origin of the multiple PPs for the LMS, the sectorial PPs for the LLS and for the LMS cannot be considered independent of each other, and consequently, the subdivision of the left liver in sectors following the B2000 does not appear to respect the principle of independence of the sectorial PPs postulated by professor Bismuth H. [8].

Second, if a surgeon plans a radical left medial sectorectomy, in 67% of patients, such surgical procedure may be questionable and not applicable because of the complex portal supply of S4 [7]. If the section of the LPV would be performed upstream from all the PPs supplying the LMS, the surgical procedure would result in a left hemi-hepatectomy; if the section of the LPV would be performed downstream from the origin of the PP for the LLS, the surgical procedure would remove only S3 and S4b, resulting in a left anterior sectorectomy (Figure 4). Consistently, to the best of authors’ knowledge, there are no reported cases of “left medial sectorectomy” or “left paramedian sectorectomy” with section of sectorial PP (the LPV) in literature.

The main advantage of this novel type of liver resection is the possibility to perform an anatomical radical liver resection in patients with tumors invading the distal part of the UPLPV, minimizing the volume of resected liver and avoiding a major resection. As the LAS is detectable in 98% of patients, the left anterior sectorectomy can be theoretically performed in the same percentage of patients, in particular in all patients with pattern I, II, or III of LPV’s branching following the classification previously reported [7]. The number of segments spared after the left anterior sectorectomy may vary depending on the pattern of LPV’s branching of the single patient; the maximum benefit of such procedure can be offered to pattern I patients (who can spare S2 and S4a); meanwhile, the benefit is more limited in patients in pattern II (who can spare only S4a) or pattern III (only S2). In the present series, all patients were pattern I.

The possibility to perform the surgical procedure of left anterior sectorectomy is strictly connected to the peculiar anatomy of the LPV that (for reasons connected with embryological development and fetal life) is a long conduit that runs to the periphery, yielding its segmental and subsegmental branches directly from the main trunk. Because of such anatomy and branching of the LPV, a liver tumor invading the LPV may be peripheral and can be treated with a peripheral limited resection as left anterior sectorectomy.

One more point of discussion is related to the management of the arterial and biliary aspects of the left liver during the described procedure. The concepts of LAS and of left anterior sectorectomy are both based on the portal branching, but in the left liver, the artero-biliary branching does not overlap the portal branching. Indeed, even in the B2000, the systematization in anatomo/functional units within the left liver differs from the one of the right, as demonstrated by the need for the left liver to detail the second-order division into two classifications (one based on bile duct and hepatic artery and one on portal vein) and the consequent discrepancy in the left liver between the notions of sector and section [13]. In literature, the arterial and biliary anatomy of the left liver showed a great level of complexity and variability [16,17,18]. As for the portal branching, anatomy of S4 plays a key role even for the arterial/biliary branching in the perspective to guarantee regular function in the left remnant liver after the removal of the LAS. Some authors reported that the biliary elements of S4 usually coalesced as a single duct at the angle of the LPV (formed by the UPLPV and the TPLPV); this anatomical aspect is more proximal than the left anterior plane (that is located at the level of the distal part of the UPLPV). Moreover, S4a had, in 88.89% of cases, one or more proper, deeply situated sheaths and their bile ducts mostly joined in the left hepatic duct [16]. Regarding the arterial anatomy, several authors reported that, in 67–79% of cases (and independently from the presence or not of the middle hepatic artery), the posterior portion of S4 was mostly supplied by the right hepatic artery and the anterior portion by the left hepatic artery [17,18]. In this sense, it should be considered that the procedure of left anterior sectorectomy is a peripheral resection, and the left anterior plane should transect the arterial and biliary branches at a segmental/subsegmental peripheral level. Consistently, in this limited series of three patients who underwent left anterior sectorectomy, no patient had post-operative complications, and in particular, no patient had arterial or biliary complications. These data support the idea that the left posterior segments have mostly a proper (more proximal or independent) vascular supply and biliary drainage; indeed, these data seem to strengthen the rationale of the subdivision of the left liver in sectors in terms of anterior/posterior as anatomically faithful.

In spite of that, before planning a left anterior sectorectomy and during the surgical procedure, biliary and arterial anatomy (as portal anatomy) must be accurately studied and kept constantly under control in order to assure proper inflow and biliary drainage of left posterior segments after removal of LAS.

The introduction of the concept of LAS and of the surgical procedure of left anterior sectorectomy are a demonstration of the key-role of the imaging in liver surgery in terms of anatomical investigation and pre/intra-operative guidance.

The small number of patients represent the main limitation of this study; these preliminary results need to be confirmed in larger studies.

In conclusion, in the presence of tumors invading the distal part of the UPLPV, left anterior sectorectomy could be considered as a new surgical option that is more conservative than the traditional left hepatectomy; this new surgical procedure offered, in this preliminary series, encouraging results in terms of safety and local oncological radicality. This study may represent the first preliminary demonstration supporting the rationale of the theoretical concept of LAS in the clinical/surgical practice.

## Figures and Tables

**Figure 1 diagnostics-12-00545-f001:**
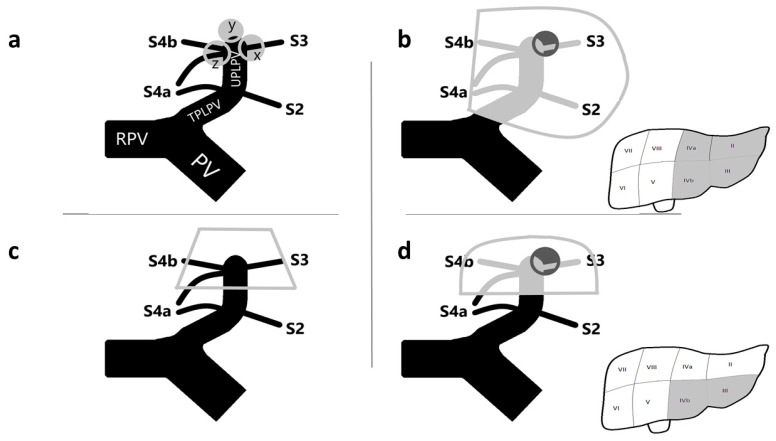
Surgical management of tumors invading the distal part of the UPLPV. (**a**) Different types of tumors invading the UPLPV; “y” invades the tip of the UPLPV; “x” and “z” invade the UPLPV and the origin of the portal pedicles for S3 and S4b, respectively. (**b**) A schematic illustration of the left hemi-hepatectomy for a tumor invading the UPLPV at the origin of the portal pedicle for S3; the portal branches (bigger picture on the left) and the portion of parenchima (smaller picture on the right) to be removed during left hepatectomy are represented in grey. (**c**) The grey line delimits the LAS, an independent anatomo/functional unit corresponding to the anterior portion of the left hemiliver including S3 and S4b inferior; it is supplied by an independent sectorial portal pedicle represented by the distal part of the UPLPV downstream from the origin of independent portal pedicles for left posterior segments (S2 and S4a). (**d**) A schematic illustration of the left anterior sectorectomy for a tumor invading the UPLPV at the origin of the portal pedicle for S3; the portal branches (bigger picture on the left) and the portion of parenchima (smaller picture on the right) to be removed during left anterior sectorectomy are represented in grey. (PV, portal vein; RPV, right portal vein; TPLPV, transverse portion of the left portal vein; UPLPV, umbilical portion of the left portal vein; LAS, left anterior sector; S2, segment 2; S3, segment 3; S4a, segment 4 superior; S4b, segment 4 inferior).

**Figure 2 diagnostics-12-00545-f002:**
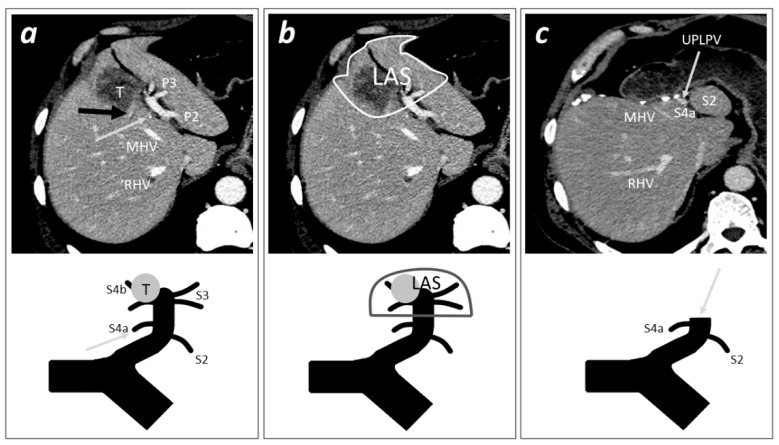
A case of tumor invading the distal part of the UPLPV treated with Left Anterior Sectorectomy. The careful pre-operative evaluation of CT-scan is fundamental to identify the pattern of LPV’s branching, to ascertain the presence of LAS, and to indicate the surgical procedure of left anterior sectorectomy. Each picture (“a”, ”b”, and “c”) includes a CT-scan imaging (above) and a schematic illustration of the portal skeleton (below) of the pre- or post-operative hepatic portal anatomy of a patient with a tumor invading the distal part of the UPLPV and the right horn of the LPV. (**a**) (Pre-operative) the CT-scan shows a portal pedicle represented by one branch for S2 (P2); a portal pedicle with two branches for S3 (P3); a portal pedicle originating from the right horn, including one branch for S4b (not visible, completely invaded by the tumor) and one branch for S4a invaded by the tumor at its origin (black arrow); and an independent portal pedicle for S4a originating from the UPLPV (grey arrow). (**b**) (Pre-operative) The patient can be classified as pattern I of LPV’s branching (presence of at least one independent portal pedicle for each of S2, S3, S4a, and S4b) [7]; the delimitation of the LAS, including the distal part of the UPLPV, the right and left horns, S3, and S4b; when planning a left anterior sectorectomy, the section of the UPLPV must be planned with the goal to remove the distal branching (left and right horns) and to spare the portal pedicles for the posterior segments (S2 and S4a), with particular attention to the origin of proximal independent portal pedicles for S4a. (**c**) (Post-operative) After the left anterior sectorectomy, the grey arrow indicates the stump of the LPV sectioned. (T, tumor; P2, portal pedicle for segment 2; P3, portal pedicle for segment 3; MHV, middle hepatic vein; RHV, right hepatic vein; S2, segment 2; S3, segment 3; S4a, segment 4 superior; S4b, segment 4 inferior; LAS, left anterior sector; UPLPV, umbilical portion of the left portal vein).

**Figure 3 diagnostics-12-00545-f003:**
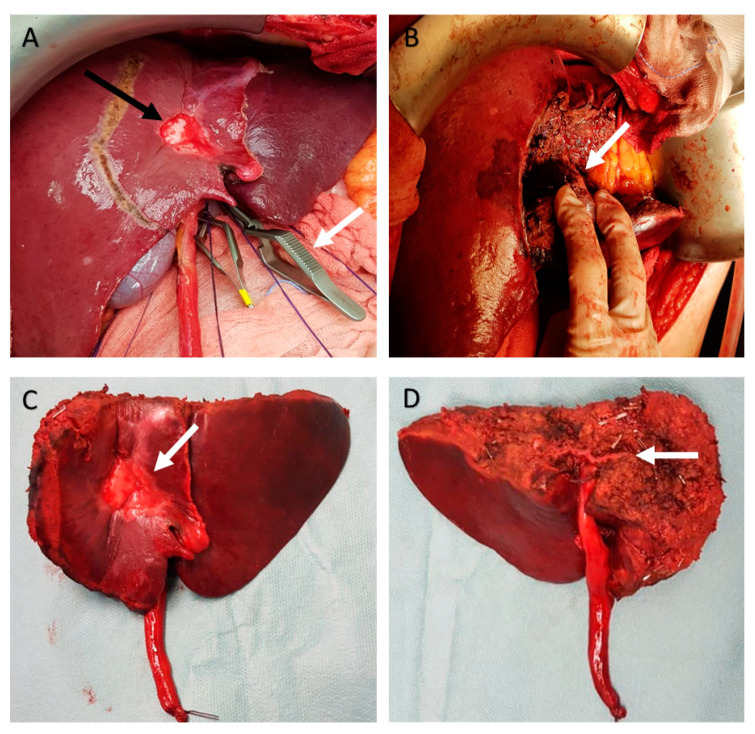
Intra-operative and specimen pictures. This figure illustrates the case of patient with a tumor invading the distal part of the UPLPV and the right horn of the LPV; the patient underwent left anterior sectorectomy. (**A**) The delimitation of the LAS (the planned transection line was drawn with electrocautery on the liver surface); the right boundary of the LAS is represented by the middle hepatic vein that was identified with ultrasound-guided technique; the intersegmental plane between S2 and S3 was identified occluding the origin of the portal pedicle for S3 with a bulldog clamp (white arrow) with consequent discoloration of S3; for the identification of the intersegmental plane between S4b and S4a, the occlusion of the portal pedicle at the right horn was not practicable because of tumor’s invasion (black arrow), so it was identified with ultrasound-guided technique in a plane between the tumor and the origin and course of proximal independent portal pedicle for S4a. (**B**) The parenchymal transection was accomplished without sectioning any segmental portal pedicle until the skeletonization of (white arrow) was obtained; the sectorial Glissonean pedicle of LAS includes the UPLPV and the peripheral arterial and biliary branches for S3 and S4b. (**C**) The specimen of left anterior sectorectomy (anterior; white arrow indicates the tumor). (**D**) The specimen of left anterior sectorectomy (posterior; white arrow indicates the stapled stump of the sectorial Glissonean pedicle of LAS).

**Figure 4 diagnostics-12-00545-f004:**
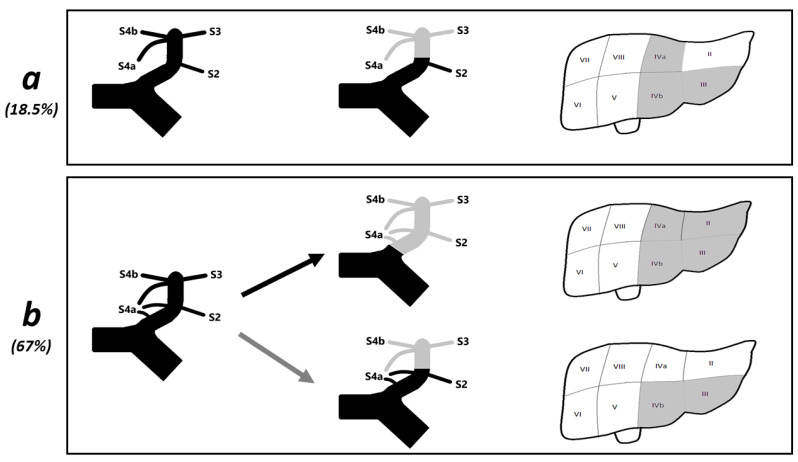
Implications of the left portal branching on the sub-division in anatomo-functional unites of the left liver. A schematic illustration of the left medial sector (LMS). The systematization of the left liver in sectors following the Brisbane Classification was applied in two different patterns of branching of the LPV (pictures on the left), with particular attention to the identification of the sectorial portal pedicle supplying the LMS (colored in grey in the picture in the middle) and the portion of parenchima corresponding to the LMS (colored in grey in the picture on the right). (**a**) This picture shows a pattern of left portal branching in which both S4a and S4b are exclusively supplied by multiple portal branches originating from the right horn; no independent portal pedicles for S4a are present (picture on the left). The sectorial portal pedicle (picture in the middle) and the portion of parenchima (picture on the right) corresponding to LMS are represented in grey. In this this type of branching of the LPV, the UPLPV (downstream from the origin of the portal pedicle supplying S2) clearly represents the sectorial portal pedicle for the LMS. This type of branching of the LPV corresponds to pattern III and is present only in 18.5% of patients [7]. (**b**) This picture shows a pattern of left portal branching in which S4a and S4b are supplied by a common portal pedicle originating from the right horn; additionally, S4a is supplied by two independent portal pedicles originating from the angle of the LPV and the TPLPV (picture on the left). This type of branching of the LPV corresponds to a subset of pattern I and is present in 67% of patients [7]. In this this type of branching of the LPV, the sectorial portal pedicle of the LMS can be identified following different criteria: Black arrow: considering that the entire S4 should be considered part of the LMS, the origin of the sectorial portal pedicle for the LMS should be identified upstream from all portal pedicle supplying S4. As a consequence, the segmental portal pedicle for S2 would be included in the sectorial portal pedicle for the LMS, and the LMS would correspond to the left liver. Grey arrow: considering that S2 should be excluded from the LMS, the origin of the sectorial portal pedicle for the LMS should be identified downstream from the origin of the portal pedicle for S2. As a consequence, the independent proximal portal pedicles for S4a would be excluded by the sectorial portal pedicle for the LMS, and the LMS would correspond to the LAS. (S2, segment 2; S3, segment 3; S4a, segment 4 superior; S4b, segment 4 inferior).

**Table 1 diagnostics-12-00545-t001:** Patients’ characteristics and post-operative results.

	Patient 1	Patient 2	Patient 3
Age (at surgery)	63	72	56
Gender	M	F	F
Histology	CRLM	CRLM	NETLM
Number of lesions	7	7	2
Site of lesions	Bilobar	Bilobar	Left hemiliver
LPV’s branching pattern	I	I	I
Pre-operative radiological presence of LAS	Yes	Yes	Yes
Surgical procedure	Left anterior sectorectomy + multiple metastasectomy	Left anterior sectorectomy + multiple metastasectomy	Left anterior sectorectomy
Cholecystectomy	Yes	Yes	Yes
Pringle time (min)	0	35	0
Operative time (min)	515	440	223
Oncological radicality	R0	R0	R0
Morbidity	-	-	-
90-day mortality	-	-	-
Blood loss (mL)	300	500	250
Follow up (months)	28 (alive)	22 (alive)	19 (alive)
Cancer recurrence	Yes	Yes	Yes
Liver recurrence	Yes	Yes	No
Cut-edge recurrence	No	No	No

## Data Availability

The data presented in this study are available on request from the corresponding author.

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
