# Peer review of "Left Anterior Sectorectomy: An Alternative to Left Hepatectomy for Tumors Invading the Distal Part of the Left Portal Vein"

_diagnostics, 2022, doi:10.3390/diagnostics12020545_

Round 1
Reviewer 1 Report
The work is original and well presented. The major problem is the very little number of patients, that it's also in contrast with your statement "as the LAS is detectable in 98% of patients, the left anterior sectorectomy could be performed in the same percentage of patients". So maybe you have to better explain why of the 92 patients who underwent liver resection only 3 underwent LAS (maybe because i.e. all the other patients have lesions located only in the right liver?).
Line 64 and 215: replace "underwent to" with "underwent"
Line 65 and 342: replace "undergone to" with "undergone"
Line 302-304: replace“the possibility to perform in patients with tumors invading the distal part of the UPLPV an anatomical radical liver resection” with “the possibility to perform an anatomical radical liver resection in patients with tumors invading the distal part of the UPLPV”
Author Response
Thank you for your comments. I agree that the number of patients involved in the study was limited. I have modified the statement "as the LAS is detectable in 98% of patients, the left anterior sectorectomy could be performed in the same percentage of patients" as follow: "as the LAS is detectable in 98% of patients, the left anterior sectorectomy could be theoretically performed in the same percentage of patients" (line 305). The first part of this statement refers to a previous study (Garancini M, et al. Branching patterns of the left portal vein and con-sequent implications in liver surgery: The left anterior sector. Hepatobiliary Pancreat Dis Int. 2021;7:S1499-3872(21)00121-1) concerning the radiological identification of the LAS in a cohort of patients; as you commented, the possibility to perform the surgical procedure of left anterior sectorectomy in 98% of patients could only be considered a potential consequence.
As reported in the ”Material and Methods” section, the indication for a left anterior sectorectomy is the presence of a tumor invading the left anterior sector invading the distal part of the umbilical portion of the left portal vein (a relatively rare indication, this part was specified in lines 64-65). The explanation for the application of this surgical procedure in only 3/92 patients was the presence of this complex relationship between the tumor and the left portal vein only in 3 among 92 patients.
The manuscript was modified following the indications of the reviewer (line 64 and 215: "underwent to" was replaced with "underwent"; line 65 and 342: "undergone to" was replaced with "undergone"; line 302-304: the sentence was modified (modifications were highlighted).
Reviewer 2 Report
thank you for the opportunity to review the paper about limited anatomical resection for clear margins in the left liver to spare a formal left hepatectomy in suitable cases.
The implication of this technique is the need for adequate preoperative imaging studies before proceeding with surgery.
What was the usual imaging modalities used for preoperative imaging and was there any usage of 3d anatomical reconstruction software in the authors institution? if yes, please detail in the manuscript.
did the authors consider the use of ICG in their surgeries for anatomical demarcation of the cut line to complement the use of intraoperative u/s?
Author Response
Thank you for your comments.
The usual imaging modalities used for preoperative evaluation before the surgical procedure were CT scan and MRI; intra-operatively, intra-operative ultrasound was routinely used. 3D anatomical reconstruction software was never used for this study.
The ICG imaging was not available in our institution during this study.
Both 3D anatomical reconstruction software and ICG imaging could be useful tools during this procedure; if available, they will surely be considered in the future.